# Christian Citizens in a Democratic State: Is a True Separation of Church and State Really Possible?

**David Haines**

Bethlehem College & Seminary, Minneapolis, MN 55415, USA; david.haines@bcsmn.edu

**Abstract:** In many North American Protestant circles, especially those with Baptist or Free Church roots, the notion of the total separation of church and state is presented as the ideal to be attained in all church and state relations. We are told that the state should have no legislative power to ordain anything in relation to church doctrine or practice, and that the church should be entirely excluded from all political, secular, or state actions. In this paper, we are going to suggest that such an approach to church–state relations (even though some might think that it flows from or is necessary for democracy) is, in fact, impossible in a true democracy. We will first consider the nature of the church and the state, and present three principles that Maritain suggests are first principles in this debate. We will then look at the classical notion of the "Citizen". We will conclude by arguing that based upon the nature of a citizen, of the church, and of the state, a strict separation of church and state is, in fact, impossible.

**Keywords:** church–state relations; democracy; citizenship; political theology; separationism

## 1. Introduction

To say that the relation between the church (or institutionalized religion) and the state is a controversial question would be to state the obvious.[1] In many North American Protestant circles, especially those with Baptist or Free Church roots, the notion of the total or absolute separation of church and state is presented as the ideal to be attained in all church and state relations. We are told that the state should have no legislative power to ordain anything in relation to church doctrine or practice and that the church should be entirely excluded from all political, secular, or state actions. The state should have no power over church doctrine and practice, and the church should have no power over the establishment of state laws or policies. Early English Puritans, such as Roger Williams (c. 1603–1683), and early American Baptists, such as Isaac Backus (1724–1806) (who was influenced by the likes of Williams and the English philosopher John Locke (1632–1704)), argued energetically for religious freedom. It appears that it was in part due to the influence of certain American Baptists—in the discussions concerning the relation of church and state and the importance of religious freedom—that Thomas Jefferson wrote to the Danbury Baptists in an attempt to articulate what would become known as "the wall of separation" between church and state—an expression that has been taken to express an absolute (or almost absolute) separation of church and state.[2]

There are a variety of views about the actual meaning of the "wall of separation" doctrine found in Jefferson's letter.[3] As we cannot hope to interact with them all, here, for the purposes of this paper, we will take the interpretation given by Derek H. Davis, who argues that Jefferson took great care in composing the letter in which this phrase was coined.[4] Davis suggests, contrary to some readings of Jefferson's letter, that "The charge that Jefferson's 'wall of separation between church and state' was one-directional only, that is, that the 'wall' was to protect the church from government but not the government from the church, is insupportable".[5] Rather, says Davis, "Jefferson was undoubtedly 'two-directional' in his view that government should have no role in advancing or promoting

religious ideas".[6] Davis summarizes Jefferson's "wall of separation" position as follows: "As we examine Jefferson's full record, it is apparent that he believed that religion and government both benefit if they maintain a healthy distance from each other. He believed that religion almost always exists in greater purity without the support of government, that only voluntary faith is authentic, and that government nurture destroys true religion"[7] The doctrine of the wall of separation, in relation to church–state relations, is taken by many, therefore, to say that religions should not put their nose into things that pertain to the state and vice versa.[8] This approach to church–state relations seems to assume that an absolute separation (all reasons for state policy and law making must be non-religious and secular, and the state must not mandate anything in relation to religious doctrines or practice) is the best way to maintain two fundamental principles that are taken to be important for the common good, i.e., the liberty of conscience and the liberty of assembly.

In this paper, we are going to suggest that such an approach to church–state relations (even though some might think that it flows from or is necessary for democracy) is virtually impossible, without major concessions, in a true democracy.[9] To prove this conclusion, we will first consider the nature of the church and the state, presenting three principles that Jacques Maritain suggests are first principles in this debate.[10] We will then look at the classical notion of the "citizen". We will conclude by arguing that, based upon the nature of a citizen, of the church, and of the state, a strict separation of church and state is, in fact, impossible in a democratic state. It is worth noting that we are neither arguing for or against democracy nor for or against any particular form of democracy. Our argument concerns the possibility of the absolute separation of church and state in a democratic state.

## 2. Church and State

Maritain begins his work *L'homme et l'état* with the following comment about the attempt to define the terms *nation*, *political body*, and *state*: "There is no task more unsat-isfying [*ingrate*] than to attempt to distinguish and circumscribe in a rational manner, in other words, to attempt to raise to a scientific and philosophical level, the most common notions which are born of the practical and contingent needs of human history and which are filled with social, cultural, and historical implications which are as ambiguous as they are fertile, yet which however encompass a center of intelligible meaning. They are no-madic concepts, unfixed; they are mutable and fluid, employed sometimes as synonyms, sometimes as contraries".[11] The truth of such a comment, in light of the history of political philosophy and theology, can very well be discouraging. However, it is necessary as we take the first step in our argument that we posit some definition of our key terms, that is, the two "political bodies" known as "church" and "state". Thus, we will look for help in the writings of Aristotle, Aquinas, and two of their modern interpreters, Jacques Maritain and Jacques Leclercq.

First of all, a "political body" can be defined, according to Aristotle, as follows: "Every state [the Greek word translated 'state' here, could also be translated 'political body' or 'political community'] (Aquinas 2007, p. 4) is a community of some kind, and every community is established with a view to some good".[12] The most basic and fundamental part of the state, out of which it grows, is the union of man and woman, called the family, by which the human race is propagated.[13] Aristotle goes on to provide what might be called an observational description of how states are formed: groups of families come together into a village,[14] and several villages unite to form a single community known as a state.[15] A *political body* or *state*, is an association of humans, established by humans, for the sake of some common good.[16] The state is, for Aristotle, the highest form of human society, and is prior to the others (as a whole is prior to the parts).[17] Aristotle will argue that it is natural for man to form communities.[18] If we were to stop here and compare Aristotle's definition of political body with the biblical understanding of the church, we might say that the church is, in a sense, a political body, i.e., it is an association of humans that is directed towards a common good. However, there are at least two key differences here. First of all, the church is not an association of humans that is established *by* humans. Secondly, it

is directed not to a common temporal good, but, according to Christian doctrine, to the ultimate good of humanity—God.

Turning to Jacques Maritain's approach to these concepts, we find him distinguishing between the notions of community, society, nation, political body, and state. He notes, first of all, that *social life* brings men together in the pursuit of a common goal.[19] This presents itself in a variety of ways in different associations of human persons, allowing for a distinction between these terms. Maritain suggests that the *community* and the *society* can be distinguished as follows:[20]

| Community | Society |
|---|---|
| The object pursued precedes determinations of the intellect. | The object pursued is determined by the intellect. |
| Produced by hereditary, linguistic, ethnical, and regional pressures. | Produced by reason. |
| Proceeds from historical and situational contexts that favor the group, and, thus, are from nature. | Proceeds from human desires, in favor of personal interests, and, thus, from human free will. |
| Constraints are based upon nature. | Constraints are based upon positive laws or ideas related to the common goal. |

A *community*, then, could be defined as a natural grouping of individuals produced by a common historico-geographical and, sometimes, ethnical background with a common goal, whereas a *society* is a voluntary association of individuals produced from agreement about common ends and held together by rationally determining commonly accepted rules or laws. Having made these distinctions, Maritain suggests, next, that a *nation* "is a community, not a society".[21] It is an ethnical community that "becomes conscious of itself such as history has made them, which is attached to the treasure of its past and which loves itself as it knows or imagines itself to be, with a sort of inevitable introversion".[22] A nation, says Maritain, has or had a land,[23] a language,[24] its own institutions,[25] its own rights,[26] and a historical calling.[27] It is a "community of communities", but it is neither a society[28] nor a state.[29] A nation is a natural association based upon a common origin and, therefore, is a form of community as defined above.

The *political body* and the *state*, however, are both forms of society, and are related to each other.[30] The political body, says Maritain, is as the whole, of which the state is the dominating part.[31] The political body, defined by Maritain, very much resembles Aristotle's definition found in his *Politics*, i.e., "It is a reality which is concretely and entirely human, which tends towards a good which is concretely and entirely human, the common good".[32] In contrast to the political body, suggests Maritain, the state "is only that part of the political body whose special object is to maintain the law, to promote common prosperity and public order, and to manage public affairs. The State is a part *specialized* in the interests of the *whole*".[33] Contra the Hegelian understanding of the State, Maritain suggests that "man is in no way for the State. The State is for man".[34]

Leclercq does not appear to make the same distinctions as Maritain, but his definition of a state seems to agree, in general, with that proposed by Maritain. Leclercq suggests that "In its primary and proper sense, the State is therefore *the organized human association, with the specific character of an organization*".[35] In further agreement with Maritain, Leclercq, summarizing arguments he made in the first volume of this work, notes that "*the individual is the end of the State*, but *the State is not the end of the individual*; it is not, for the individual, but a means".[36] Leclercq also notes a distinction between what Maritain calls the political body and the state, though with different terms. What Maritain calls the political body seems to be what Leclerq refers to as the *state*, i.e., the organized association of individuals seeking a common good.[37] What Maritain calls the *state*, Leclercq calls the government, saying that "When the State acts, it is represented by certain men, the governors, who together form that which, in modern legal terms, we call *the government*".[38] Leclercq notes, however, that

we often use the term state equivocally to refer to both the *political body*, composed of all the citizens of a country, and to refer to the governing body, which we could designate as the *politicians*.[39] These politicians or governors, in a democratic society, are for Leclercq "simply the organs of the State, and they do not represent it except insomuch as they have received a mission".[40]

We may summarize the distinctions made by Maritain and Leclercq, for our purposes, as follows: the *state* or *political body* is the entire group of individuals who are voluntarily united in seeking a common good, agreeing upon the creation, upholding, and enforcing of specific laws to be set over the members of the state. The *governing body* or *politician(s)* may vary from a single individual to a group of individuals, and they may be the entire state or political body or only a part of the state or political body, depending upon the form of government in place.

The *church* is a form of *political body*. That is, it is a group of individuals voluntarily united in the pursuit of a common good, agreeing upon a set of laws by which they are governed. The church, of course, is more than a simple political body. Maritain helpfully distinguishes between how an unbeliever and a believer understand the church. The *unbeliever* sees churches as no more than "organized bodies or associations which are especially dedicated to the religious needs and beliefs of a certain number of his fellow travelers on earth [compagnons de route ici-bas], that is to say, to the spiritual values to which they have committed their lives and upon which their moral ideals are based".[41] To the *believer*, however,

> "the Church is a supernatural society, simultaneously divine and human. . .which unites humans within itself as equal citizens of the Kingdom of God and directs them to eternal life, which is already begun on earth; which teaches them the revealed truths received in a deposit from the incarnate Word himself; and which is the very body of which Christ is the head, a body which is visible by its very nature and as it is ontologically one, visible in the faith which he professes, in its worship, its discipline and its sacraments, and in the refraction of his supernatural personality through its structure and human activity—invisible body in the mystery of the divine grace and charity vivifying souls. . .For the believer, the Church is the body of Christ supernaturally formed from the human race, where, as Bossuet said, *the Christ is spread and communicated*".[42]

It seems to follow that the Christian may be at once a citizen of the heavenly political body known as the church and of an earthly political body or state. Maritain notes that this fact implies three key principles. *First of all*, even based upon the unbelievers' understanding of the church, Christians must necessarily be accorded both freedom of association and freedom of conscience (to believe as they will).[43] More importantly, if Christian believers are right (and I would agree with Maritain that they are), then the end pursued by the church is the highest possible end that can be pursued by human beings (union with God). It follows, as a *second* principle, that the church is superior in nature and in end to any and all temporal political bodies (that is, to all earthly states).[44] A *third* principle, which we will be discussing in what follows, is the necessary cooperation between church and state.[45]

To properly understand the distinction between the earthly and heavenly political bodies and to fully grasp the third principle raised by Maritain, we must answer the question of what it means to be a *citizen*? This must be related to the definitions given by political and governing bodies.

### 3. What Is a Citizen?

The classical definition of the term "citizen" can be traced back to Aristotle's discussion of the citizen in his *Politics*, where he says that "[h]e who has the power to take part in the deliberative or judicial administration of any state is said by us to be a citizen of that state; and, speaking generally, a state is a body of citizens sufficing for the purposes of life".[46] Commenting on Aristotle's definition of a citizen, Aquinas says that "he says first that nothing else can better define what a citizen is absolutely than the fact that one participates

in the courts of the political community, namely, can decide cases about some matters, and in the ruling power of the political community, namely, has some power in its affairs".[47] The *citizen*, then, is the individual member of a state who has some juridical, administrative, and/or determinative role or responsibility, the ability to participate, and the power to act or influence, in relation to the governance of the state and its people. This seems to entail that a citizen is a member (the only member or one of the members) of the *governing body* of the *political body*.

Aquinas goes on to note that there are different types of office in the state, e.g., those that are for a fixed term and those that are not so limited.[48] As to the "fixed term office", the person holding this office is only allowed to hold the office for a limited amount of time or for a limited number of times.[49] In relation to the office for which there is no fixed term, the individual may occupy this office for as long as they want or for as many consecutive terms as they so desire.[50] Such types of offices may include sitting on a jury, being a member of the governing body, and so on.[51] Any person who may rightfully occupy either type of office in a particular state, suggests Aquinas, would be considered a citizen.[52]

Who the citizen is, or who the citizens are, will differ, suggests Aristotle, based upon the type of state being discussed. In other words, if the citizen is a person who can rightfully occupy an office in the state, then, using the types of true states described in Aristotle's *Politics*,[53] it becomes clear that, in a monarchy, there is only one citizen, the monarch; in an aristocracy, only the ruling elites (by their excellent virtue) are citizens; and, in many versions of democracies or republics,[54] all those who are born within the geographical limits of the state, or who successfully purchase citizenship, or who may legally bear arms, and who are considered free, are citizens.[55] The *governing body*, then, is the sum total of all citizens of a given *political body*.[56] In a democracy or representative republic, then, the people are the governing body; the group of "politicians", if not the sum total of the people, are but civil servants and representatives of the people.

To be a citizen of the kingdom of heaven, put as simply as possible, is, as we saw above, to be in Christ. Those who are "born" by faith into the church are citizens of heaven. This is true of the Christian, whether or not they are members of a local assembly of believers (though Christians do tend towards assembly with other believers for the sake of worship, mutual encouragement, and participation in the sacraments). Christians could be said to have, therefore, dual citizenship, i.e., earthly and heavenly. Local and national churches often have similar political (governing) structures as those mentioned above in relation to the earthly state, e.g., a single pastor or spiritual leader (monarch-like leadership), a group or board of pastors or spiritual leaders (aristocracy-like leadership), or congregationalism (social democratic- or republic-style leadership).

When discussing the question of church and state, one of the difficulties we run into could be called *the problem of extension*. When we are talking about "the church" determining how the state must act or the state determining how the church must act, are we to understand the church as the universal and invisible body of Christ, as a national denomination, as a local body of believers, or as a single believer? Also, how do we understand the state? As the single leader? As the leading party? As the "will of the people"? For example, in a strict monarchy, there is only one citizen, the monarch. If the monarch is a Christian and rules the state according to Christian principles, would this be sufficient for us to say that the church is governing the state. If so, then it would not be necessary for a whole local church, national association, or international body of believers to be actively engaged in politics for us to say that the church is governing the state. We suggest, therefore, that to discuss the question of the directive influence of church over state or vice versa, we need only assume that a portion of those considered to be citizens of the state (as defined above) are influential in the church or that only a portion of the members of any one church (or of one religious group) are influential in the state.[57]

### 4. The Great Divorce: An Impossible Separation

Bringing together the observations made in the preceding sections, it seems clear that in any temporal and earthly state there will be an overlap of church and state that is so significant as to make an absolute separation of church and state impossible. There are, we suggest, two primary ways in which this overlap is both inevitable and reasonable, making, therefore, an absolute separation of church and state strictly impossible.

#### 4.1. Overlapping Origins, Ends, and Means

One way in which we can discern overlap, is by pointing to overlapping origins, ends, and means of church and state.[58] The first way in which there is an overlap of church and state is in relation to their origins. Without affirming the "divine right of kings", Christian scriptures affirm that both the temporal states and the heavenly state originate in the divine will. In relationship to the temporal states, God made human nature, and, as Aristotle suggests, man is by nature a political animal.[59] Secondly, according to Christian scriptures,[60] God providentially gives the temporal sword to the temporal governing body.[61] As such, according to Christian scriptures, temporal states originate in the divine will. We have already noted that the church, understood both in its local instances and in its universal body, is a divine institution, with a divine head originating in the divine will. This aspect of Christian doctrine has led some theologians to affirm that the leader(s) of the temporal state must always pursue the glory of God (Calvin 2006, pp. 11–12). This would, of course, destroy any true separation of church and state.

A second way in which there is a necessary overlap of church and state, based upon our definitions, concerns the respective ends (and means to obtain the ends) of both the earthly states and the heavenly state. Both earthly states and the heavenly state seek to direct their members to the proper good of mankind (both temporal goods and the ultimate good). Assuming a somewhat virtuous governing body, the earthly state directs its citizens towards civil virtues and their earthly temporal common goods; the heavenly state, though it may be generally understood to be directing men towards divine virtues and union with God, also directs its members towards temporal goods and virtues. As Maritain notes, the end of the church is necessarily higher than the end of any one (or all) earthly states, as it is the ultimate good towards which all humans, as human, tend. As such, it may plausibly be argued that the state should follow the church's guidance in directing its members towards virtuous living within the world. On the other hand, the church teaches its members that to be a virtuous member of an earthly state (by fully accomplishing all of the responsibilities of the office a person finds themselves in, which may mean no more than paying taxes, but may also include voting, making legal decisions as a judge, sitting on a jury, or even governing the state) is a required duty of all those who are members of the heavenly state.[62] We find, here, a key area of necessary overlap as follows: the church requires that its members obey the earthly state insomuch as the earthly state does not require them to disobey the church. However, if the earthly state requires the members of the church to disobey their spiritual sovereign—God—then the earthly state is to be disobeyed or its unjust ordinances ignored. Indeed, any positive law that is necessarily opposed to natural law is no law. The overlap, then, may involve conflict (as when the earthly state requires actions that cause its members to act contrary to the ordinances of the heavenly state, or when the members of the church acting as earthly citizens call out the immorality of the state and its laws or call for justice to be carried out, or when the state-financed education system teaches its members to believe or think in ways that contradict church teachings), or happy cooperation (as when both state and church agree on ends to be pursued, such as justice, helping the poor, teaching true moral principles in state-financed education, and so on).

In relation to the laws governing the two institutions, positive law and the *jus gentium* are the means by which the earthly states teach their members their common end and how to attain it. Divine law, ecclesial laws, and the scriptures are the means by which the heavenly state teaches its members their common end and how to attain it. Discipline

(physical or spiritual) is used by each state to enforce these laws. Once again, there will be overlap here. For example, if the earthly state requires its members to approve or even engage in forms of abortion, then members of the heavenly city are required to follow the higher laws of the church, which prohibit abortion (either the approving of it or engaging in it). Other areas of contention may include the public preaching of the gospel, the publishing and propagating of the word of God, or agreement upon and participation in moral issues related to transgenderism, euthanasia, the education of children, and so on. The law teaches, and when the law (ecclesial or state) teaches moral depravity, intervention is required. The church should support and lobby for just laws, and fight ardently against unjust laws. On the other hand, if the church (or a religion) installs laws within their religious community that are unjust or evil, the state should intervene. Here, again, we have either a happy accord or a necessary conflict in relation to human actions here on earth.

Speaking generally, it could be said that the means by which the members of both earthly and heavenly states seek to attain their proper ends include both physical things and spiritual actions and practices. For example, money, possessions of all sorts (land, clothing, modes of transportation), time, physical energy, and so on are all used by members of the earthly states for the good of the state and its citizens.[63] These things are also used by the church for the good of the church and its members. Necessarily, then, there may be a happy cooperation (as when the church and the state cooperate in pursuing the well-being of all men through welfare programs) or an unfortunate conflict (as when the state impedes the meeting of church members for unjustified reasons). One of the main areas of contention, in relation to means, will be education. Citizens of the heavenly state will both be educated and provide education in state-financed institutes of education. Is the Christian teacher or professor to be required to put aside their religious convictions in relation to all subjects touching reality and morality? Is this possible?

There will be, therefore, in relation to the ends, laws, and means, a great deal of overlap between church and state. Now, one might propose that the church should never impede the state in its proper actions and the state should never impede the church in its proper actions, but even this proposal is a principle that implies interaction and cooperation between church and state. There is, however, another way in which church and state necessarily overlap, which is more obvious and unavoidable, that is, by the members of the two societies.

*4.2. Overlapping Members*

The overlap—or identity—of the members of the political bodies of church and state, and, thus, the impossibility of the separation of church and state, is most obvious and most significant in the contemporary West in the case of a democratic, socialist, or republican form of government. For the sake of this argument, we will use the basic political structures found in North American countries such as Canada and the United States. In these countries, you are a citizen if you are born in the country in question, born to at least one parent who is a citizen of the country in question, or if you are able to apply for and obtain citizenship.

If you are a citizen, then you are also (whether or not you take advantage of it) endowed with both the privileges and the responsibilities of a citizen. This means that you are a part of the governing body of individuals who have the right and responsibility to express their desires in relation to the proper government of the country (in relation to the laws and policies that are enacted and how they are enacted, in relation to the ministries or government organizations that are formed and how they function, and in relation to how the government spends its money), you have the right to run for any office and occupy any office won (including mayor, governor, ministerial positions, or even the president or prime minister), and you have the right and responsibility to let those who currently hold public offices know (through phone calls, letters, manifestations, public presentations, and so on) how you think they should be using their time, state money, and the representational power given to them by the popular vote. You may be required to serve on a jury, you may

need to act for the common good as a volunteer deputy or army officer, and you may need to put a dangerous individual under citizen's arrest.

In each of the actions thus discussed, and many more, the individual is acting according to the rights and responsibilities with which they are endowed as citizens of an earthly state. When the citizen in question happens to be a Christian (a member of the heavenly state), it is as a member of the church that they are acting. When a Christian votes, they are expressing their opinions concerning the subject voted upon (perhaps the voting in of a government official, the enacting of a law or policy, or whatever else is being voted) and how they think that the government should act for the common good, and they are performing this *as a Christian*. There is no separation of church and state here. The individual members of the visible political body called the church, when they vote (or abstain from voting) as citizens of a temporal state, when they sit on a jury, when they act as an official, or when they engage in any other action that benefits the common good of the earthly state in which they are citizens, they are bringing the moral and religious positions of the heavenly city to bear on the earthly city, thus actively seeking to influence the earthly city.

This point has been noted by both Maritain and Leclercq. Maritain notes, first of all, when elaborating his third principle concerning the relation between church and state, that "[f]rom the fact that the same human person is, at the same time, a member of that society which is the Church and a member of that society which is the political body, an absolute division between these two societies would signify that the human person must be cut in two".[64] This fact entails Maritain's third principle, the necessary cooperation between church and state.[65] Later, in discussing the place of the church in a democratic state, he comes to the precise conclusion for which we have argued, i.e., when the political body is democratic, the governing body is the people, neither a king nor politicians, but the people who put public servants in place by popular vote. In this case, the church, through its members, is a part of the political body. Furthermore, in its relationship with the state, it is not the "politicians" that it must deal with but the other citizens of the state, i.e., the people.[66] The church, in the democratic state, is a part of the people that constitutes the governing body of the state and, therefore, when it acts or speaks, it is as equally a part of the heavenly state as it is a part of the temporal state.

When it seeks to evangelize unbelievers in the state, it is seeking to convince other citizens (politicians and governors) of the earthly city to join the heavenly city. In so doing, the church is seeking to extend its influence over the earthly state. Indeed, suggests Maritain, the state always has an obligation towards the truths to which the political leaders adhere, but, in a democracy "the political body in itself has an obligation towards the truth to which *the people itself*, or the citizens which constitute the political body, adhere in conscience. The political body knows no other truth than that which is known by the people".[67] In other words, if the people are primarily Christian and the state is democratic, then the state has the obligation to promote that truth which is adhered to by the majority of the people—Christianity? It turns out, then, that "everything will depend, in practice, upon what the people or citizens believe in liberty of conscience".[68]

A similar argument can be found in the writings of Jacques Leclercq, as he is interacting with what he qualifies as a version of liberal democracy, which he sees as originating in the thought of philosophers such as Jean-Jacques Rousseau.[69] Leclerq notes that, in a liberal democracy, human freedom is the foundational principle of the liberal democracy. "Men, abandoning their liberty to the community rediscover it within her, as they are the community; but the community cannot, itself, remove this freedom. The people cannot call upon a sovereign; it is, itself, the sovereign, and the only sovereign possible".[70] In a democracy, the will of the people is sovereign, i.e., what the people believe will, ultimately, influence the rule of law in the state.

The implication appears to be that, if Christians, as members of the heavenly state, are living in and fulfilling their responsibilities as members of the earthly democratic state, then the church is overflowing into, influencing, even dictating and determining how the

state will function. Inversely, insomuch as the church itself also functions with a democratic form of governance, as is found in most Baptist and Free Churches or any church that can be described as "congregational", it turns out that the direction of the local church, its doctrines and practices, how it pursues its goals locally, preaches the gospel, and interacts with the surrounding society will also be influenced and even governed by citizens of the earthly democratic state. The overlap is inevitable, separation is impossible.

**5. Conclusions**

In light of the concepts and principles that we have discussed above, it seems that when the state is democratic and even a small portion of the citizens are Christians, it is impossible for there to be an absolute separation of church and state. In such a situation, the church and the state are mutually influential upon each other in a wide variety of issues; this is not only due to sharing overlapping ends, laws, and means, but also due to the fact that the members (citizens) of the church are also citizens of the state. The state is partially composed of church members, and the church is almost entirely composed of citizens of the state. The greater the percentage of citizens of a state who are Christian, the more the state will be influenced and governed by "the church".[71]

This situation is not avoided by the election of a "primary leader" (i.e., a president or prime minister), as (in North American representative democracies at least) the leaders are still beholden to follow the will of the people, expressed through voting, manifestations, and other means. A magistrate in a democratic state who does not follow the will of the people is no longer a democratic leader; they have become a monarch or a tyrant.

The situation is not changed when there is a plurality of religions or beliefs, for, as Maritain and Leclercq have rightly pointed out, in a democracy, the direction of the state is determined by the voice of the citizens. What this entails, practically, is that the most vocal citizens will end up directing the state. Those citizens will not be "neutral", but rather will adhere to beliefs about religion(s), be they adherents to a religion, skeptics, agnostics, or atheists. Inevitably, then, a democratic state will not be neutral towards religions but will necessarily promote the views about religions of the most vocal citizens. Again, there can be no separation of church and state. The nature of the church (i.e., religion or anti-religion) may not be what we typically refer to when discussing church–state relations, but necessarily a view on religion will govern the state.

Proposing a solution to the problems we have raised is beyond the scope of this paper. The goal of this paper has simply been to show that, assuming a democratic state, there is a major flaw with one of the most prominent and popular North American approaches to church–state relations (which appeals to the notion of a wall of separation and, at least, to an almost total separation of church and state). All that such a theory accomplishes is to provide a pseudo-guarantee that the state will never be influenced by the church and vice versa, all while ensuring that Christians will struggle to understand their place, role, and responsibilities in the earthly state, and that the non-Christian members of the state will bewail even the appearance of a Christian influence over the state.[72]

**Funding:** This research received no external funding.

**Institutional Review Board Statement:** Not applicable.

**Informed Consent Statement:** Not applicable.

**Data Availability Statement:** No new data were created or analyzed in this study. Data sharing is not applicable to this article.

**Conflicts of Interest:** The authors declare no conflict of interest.

**Notes**

1.  There is such a wide spectrum of views on how they should or should not be related, that one might almost suggest that there are as many perspectives on this question as there are authors writing about it. Andrew Naselli has recently published a helpful

article attempting to articulate, as clearly as possible, the entire range of approaches to the relation between Church and State (Naselli 2023).

2    (cf. Davis 2023, p. 93; Finn 2023, p. 447). It is worth noting, that though some theorists see Jefferson's notion of the "Wall of Separation" as expressing his view of an absolute separation of Church and State, others have pointed out that Jefferson was quite positive about the role of religion in the State and in State funded education (Benjamin 1969, p. 95). Benjamin goes on to note the historical origins of separationism in the European Anabaptist and Mennonite movements (Benjamin 1969, p. 96), and then explores a number of difficulties that he thinks flow from the Separationist approach to Church and State, including the development of a "club mentality" for many church attendees, pietism, and so on.

3    (cf. Davis 2003, p. 6). Davis notes that the notion of a "Wall of Separation" is actually found, prior to Jefferson, in the writings of Roger Williams (Davis 2003, p. 13). The notion of a "wall of separation" in the thought of Roger Williams is also discussed by Joseph M. Dawson (cf. Dawson 2008, pp. 677–78). Dawson appears to interpret the "wall of separation" as implying (if not expressing outright) the absolute separation of Church and State specifically in relation to their relative institutional functions. He suggests that, "[w]hat the Constitution of the United States forbids and constitutions of all the states forbid, although in different forms of expression, is the making of any law or the action of any governmental authority in pursuance of any law that involves the interlocking of the official functions of the state (or any of its agencies) with the official or institutional functions of any church. (Dawson 2008, p. 679)".

4    (Davis 2003, pp. 9–12).

5    (Davis 2003, p. 12).

6    (Davis 2003, p. 12).

7    (Davis 2003, p. 13). Davis does not seem to be suggesting that Jefferson, Williams, or others, suggested that there could be no "cooperation" or "interaction" between Church and State. Rather, in a different paper, Davis notes that "Scholars are fond of stressing that Williams was concerned about protecting the church from the state, whereas Jefferson felt the 'wall was necessary to protect the state from the church. While this wom-out distinction is generally accurate, there were far more likenesses than differences in Williams's and Jefferson's views on church-state relations. Clearly, both believed that a flexible boundary between the institutions of religion and government preserved the health and integrity of both (Davis 1999, p. 201)."

8    In this paper, we will be working with the approach outlined above, which resembles the view outlined by Robert Audi, who describes it as, "the state should not interfere with the church, and (though this is usually given lesser emphasis) the church should not interfere with the state. The separation doctrine is also intended to apply to the state in relation to religious individuals who are not affiliated with any church (Audi 1989, p. 262.)." There are other approaches to the question of Church and State, and, indeed, questions concerning how the "wall of separation" actually works out in practice. A very similar approach to view of the separation of Church and State which has been articulated above, can be found in the arguments of Kathleen M. Sullivan, writing against the position of Michael McConnell (cf. Sullivan 1992, pp. 195–223). Sullivan, in a way which tends to prove the point we are making in this paper, begins her article by noting that Roman Catholic and Protestant religious leaders of the 1980s and 90s tended to wield a great deal of political power (Sullivan 1992, p. 196), by influencing their voting parishioners. She then turns to a discussion of "The Free Exercise Clause and the Establishment Clause" to argue that the public and civil resolving of moral issues must be done on entirely secular grounds (Sullivan 1992, p. 197), that public education must be entirely secular (Sullivan 1992, pp. 199, 202), and "official agnosticism" in all government activities (Sullivan 1992, p. 206). It seems, upon consideration, that Sullivan's articulation of the separation of Church and State also falls prey to the argument we are presenting In this paper. In other words, an approach to Church and State which requires that all reasons for the establishment of laws or polity be secular (or non-religious), seems to fall prey to the argument presented in this paper. The inverse of this could also be shown to be problematic. That is, an approach to Church and State which requires that all church doctrine and practice must be entirely grounded in exclusively "religious" reasons will also fall prey to the argument presented in this paper (cf. Hooker 1969).

9    The nature of a "true Democracy" will be discussed below.

10   The thought of Maritain is particularly helpful for this discussion, as he defended the principles of Democracy, but also thought that it was possible for the Church and a secularized State to cooperate (thus suggesting that he thought that Democracy might not require an absolute separation of Church and State). We might even find, in Maritain, a form of Christian Democracy (cf. Hellman 1991, pp. 460–61, 471). For more on Maritain's views concerning the secularized state and his views on the separation of Church and State, see (Pink 2015). Pink's article is wide-ranging, but is helpful for drawing out how Maritain's general project failed due to the direction that was taken by the contemporary secularization of politics. Maritain thought it would lead to greater freedom for Religion, but, it ended in the State seeking to subsume Religion under its authority as one among a number of common goods (Pink 2015, pp. 12–13, 23–32). Pink ends up providing, using his interaction with Maritain, an interesting analysis of the problems created by the secularation of politics for the separation of Church and State. Pink's analysis does not appear to necessitate any major changes in the argument we are proposing in the article (but could be seen as ultimately arriving at similar conclusions by a different route).

11   (Maritain 1965, p. 1). My translation from the French, "Il n'est pas de tâche plus ingrate que d'essayer de distinguer et de circonscrire de façon rationnelle, en d'autres termes d'essayer d'élever jusqu'à un niveau scientifique ou philosophique, des notions banales qui sont nées des besoins pratiques et contingents de l'histoire humaine et sont chargées d'implications sociales,

culturelles et historiques aussi ambiguës que fertiles, et qui pourtant enveloppent un noyeau de signification intelligible. Ce sont là des concepts nomades, non fixés; ils sont changeants et fluides, employés tantôt comme synonymes, tantôt comme contraires". Jacques Leclercq also notes the difficulty of defining these terms, as they are given multiple meanings (cf. Leclercq 1958, p. 10).

12   (Aristotle 1941). Aristotle, *Politica*, 1252a1–2, trans. Benjamin Jowett, in *The Basic Works of Aristotle*, ed. Richard McKeon (New York: Random House, 1941), 1127. For all references to Aristotle's Politics, we will use this translation (unless otherwise mentioned), and adhere to standard citation practices for Aristotle's works.

13   Aristotle, *Politica*, 1252a26–33.

14   Aristotle, *Politica*, 1252b14–27.

15   Aristotle, *Politica*, 1252b27–30.

16   Aquinas, *CAP*, 7.

17   Aristotle, *Politica*, 1253a19–20.

18   Aristotle, *Politica*, 1252b30–1253a39.

19   Maritain, *HE*, 3.

20   Maritain, *HE*, 3–4.

21   Maritain, *HE*, 4.

22   Maritain, *HE*, 5. My translation from the French, "prennent conscience d'eux-mêmes tels que l'histoire les a faits, qui sont attachés au trésor de leur passé et qui s'aiment tels qu'ils se savent ou s'imaginent être, avec une sorte d'inévitable introversion".

23   Maritain, *HE*, 5.

24   Maritain, *HE*, 5.

25   Maritain, *HE*, 5–6.

26   Maritain, *HE*, 6.

27   Maritain, *HE*, 6.

28   Maritain, *HE*, 6.

29   Maritain, *HE*, 6–7.

30   Maritain, *HE*, 9.

31   Maritain, *HE*, 9.

32   Maritain, *HE*, 9. My translation from the French, "C'est une réalité concrètement et entièrement humaine, qui tend vers un bien concrètement et entièrement humaine, le bien commun. »

33   Maritain, *HE*, 11–12. My translation from the French, "est seulement cette partie du corps politique dont l'objet spécial est de maintenir la loi, de promouvoir la prospérité commune et l'ordre public, et d'administrer les affaires publiques. L'État est une partie *spécialisée* dans les intérêts du *tout*".

34   Maritain, *HE*, 12. My translation from the French, "l'homme n'est à aucun titre pour l'État. L'État est pour l'homme".

35   Leclercq, *EP*, 11. My translation from the French, "Au sens propre et premier, l'État est donc *la collectivité humaine organisée, avec son caractère spécifique d'organisation*. » Italics in Leclercq.

36   Leclercq, *EP*, 11. My translation from the French, "*l'individu est la fin de l'État*, mais *l'État n'est pas la fin de l'individu*; il n'est pour l'individu qu'un moyen". Italics in Leclercq.

37   Leclercq, *EP*, 12.

38   Leclercq, *EP*, 13. My translation from the French, "Quand l'État agit, il est représenté par certains hommes, les gouvernants, qui forment ensemble ce que, dans le droit moderne, on appelle *le gouvernement*". The italics in Leclercq.

39   Leclercq, *EP*, 12, 13.

40   Leclercq, *EP*, 13. My translation from the French, "simplement les organes de l'État, et ils ne le représentent que dans la mesure où ils en ont reçu mission".

41   Maritain, *HE*, 140. My translation from the French, "sont des corps organisés ou des associations qui se dédient spécialement aux besoins religieux et aux croyances religieuses d'un certain nombre de ses compagnons de route ici-bas, c'est-à-dire aux valeurs spirituelles auxquelles ils ont commis leur vie et auxquelles leur idéal moral est suspendu".

42   Maritain, *HE*, 140–41. My translation from the French, "l'Église est une société surnaturelle, à la fois divine et humaine...qui réunit en soi les hommes comme concitoyens du Royaume de Dieu et les conduit à la vie éternelle, déjà commencée ici-bas; qui leur enseigne la vérité révélée reçue en dépôt du Verbe incarné lui-même; et qui est le corps même dont le Christ est la tête, corps visible de par son essence et en tant même qu'ontologiquement un, visible dans la foi qu'il professe, dans son culte, sa discipline et ses sacrements, et dans la réfraction de sa personnalité surnaturelle à travers sa structure et son activité humaine—corps invisible dans le mystère de la grâce et de la charité divines vivifiant les âmes...Pour le croyant, l'Église est le corps du Christ surnaturellement formé de la race humaine, ou, comme l'a dit Bossuet, *le Christ répandu et communiqué*". Italics in Maritain.

43   Maritain, *HE*, 140.

44   Maritain, *HE*, 141–42. This view can be found expounded upon in the Roman Catholic notion of the distinction and cooperation of Church and State (cf. Konvitz 1949, pp. 46–48).

45  Maritain, *HE*, 142.

46  Aristotle, *Politica*, 1275b18–21. Cf. Aristotle, *Politica*, 1275a22–24.

47  Aquinas, *CAP*, 183.

48  Aquinas, *CAP*, 183.

49  Aquinas, *CAP*, 183.

50  Aquinas, *CAP*, 183.

51  Aquinas, *CAP*, 183.

52  Aquinas, *CAP*, 183.

53  Aristotle, *Politica*, 1279a23–1279b4.

54  We say "versions" of Democracy, as there are a variety of forms of Democratic State. The most basic understanding of a Democracy is "the rule of the many", or a people that is governed by itself. In this paper, we will be interacting with what is commonly called a *Liberal Democracy*, which as Marc F. Plattner has rightly noted, "is what most people mean today when they speak of democracy (Plattner 1998, p. 172). It is generally agreed that the primary criteria for a Liberal Democracy are: (1) that the citizens have a right to express their wishes concerning the direction of the country through free elections; (2) the protection of the various rights or liberties of the person; and (3) the rule of law (cf. Plattner 1998, p. 171.) Michael J. Perry only lists the first 2 of the three criteria we have named, in his article (Perry 2009, pp. 621–22). John Hellman's analysis of Maritain's political thought, as seen in letters written by Maritain to Yves Simon, shows that this is also, more or less, what Maritain understood as fundamental to a Democracy (Hellman 1991, p. 458).

There are, of course, other forms of Democracy. In his paper, Plattner mentions and briefly discusses the following variations: "illiberal democracy (Plattner 1998, p. 172)," "electoral democracies (Plattner 1998, p. 171)", Republican or Representational Democracies versus Pure Democracies (Plattner 1998, p. 174), and so on. He also suggests that for Montesquieu and Rousseau, as soon as a Democracy turns towards "representation" it is no longer truly democratic or free (Plattner 1998, p. 174). Ralph Ketcham, in his book *The Idea of Democracy in the Modern Era*, also discusses forms of Democracy such as "social democracy", or, he suggests, "the post-1945 liberal corporate state" (Ketcham 2021, pp. 90, 99) He also appears to use, at least, the first two criteria we mentioned above in affirming that some form of governance is democratic (cf. Ketcham 2021, pp. 20–22, 41, 44, 93, 99, and so on). For a discussion of other versions of Democracy, see (Miller 1984, pp. 205–8).

55  Aquinas, *CAP*, 184.

56  Aquinas, *CAP*, 184.

57  The question of religious pluralism in a given State creates even greater complications for the Church-State debate.

58  Dawson points to Roger Williams as one who would disagree with what we are suggesting in this section, suggesting that "As argued by Roger Williams in the beginning, the functions and objectives of religion and state differ. (Dawson 2008, p. 681)". We propose to show that so understood, Dawson, and Williams as interpreted by Dawson, are wrong. Rather, as we will show, Church and State overlap in relation to origins, ends, and means.

59  Aristotle, *Politica*, 1253a2.

60  Cf. Ex. 22:28; Rom. 13:1–7; 1 Tim. 2:1–2; 1 Pet. 2:13–17.

61  This can be taken to mean both: God divinely ordains that man must live in a temporal political body with some form of governing body and God providentially governs all of human political and governing bodies (raising and tearing down all human governments as He sees fit). This is, of course, a point of contention for Christian theologians.

62  Cf. Ex. 22:28; Rom. 13:1–7; 1 Cor. 7:20–24; Phil. 4:11; 1 Tim. 2:1–2; 1 Pet. 2:13–17.

63  The point of overlap we are noting here appears to be what Konvitz is pointing to when he discusses the question of monetary support (or lack of support) for religious institutions, including religious schools (Konvitz 1949, pp. 48–50). Konvitz goes on to discuss the possibility of distinction and cooperation in the United States, ultimately arguing that the wall of separation idea does not allow for the possibility of the doctrine of distinction and cooperation (Konvitz 1949, pp. 50–60).

64  Maritain, *HE*, 143. Translation from the French, "Du fait que la même personne humaine est à la fois membre de cette société qu'est l'église et membre de cette société qu'est le corps politique, une division absolue entre ces deux sociétés signifierait que la personne humaine doit être coupée en deux".

65  Maritain, *HE*, 143.

66  Maritain, *HE*, 154.

67  Maritain, *HE*, 154. Translated from the French, "le corps politique en tant que tel a une obligation à l'égard de la vérité à laquelle *le peuple lui-même*, ou les citoyens qui constituent le corps politique, adhèrent en conscience. Le corps politique ne connaît pas d'autre vérité que celle que le peuple connaît". Italics in Maritain.

68  Maritain, *HE*, 155. Translated from the French, "tout dépendra, en pratique, de ce que le peuple ou les citoyens croient librement en conscience".

69  It was common amongst philosophers of the early to mid-1900s to portray Rousseau as one of the greatest Democratic theorists of the 18th century (cf. Miller 1984, p. 2). This approach has since been questioned, creating something of a debate on the

matter. Though some might be inclined to outright deny that Jean-Jacques Rousseau can be portrayed as proning either political liberalism or Democracy (and there is certainly reason to doubt that Rousseau promoted Democracy in any way that would be recognized by Democratic theorists today. cf. Miller 1984, p. 2), there is reason to think that both of these notions can be found in Rousseau (and, thus, some primitive form of Liberal Democracy). For example, Rousseau has been portrayed as denying that a Representational Democracy is a true Democracy (Plattner 1998, p. 174), but, far from rejecting Democracy, Rousseau is generally interpreted as affirming that freedom is only possible *through* a form of Democracy (cf. Willhoite 1965, p. 501). Furthermore, not only is Rousseau taken to affirm Democracy, he also appears to have promoted the three main criteria of a Liberal Democracy that we outlined above, namely: the idea that the will of the people is the source of the constitution and law, that the rule of law is supreme over people and government (Dunning 1909, p. 406), the importance of individual liberty (Dunning 1909, p. 405), and, that his thought is the precursor for the notion of free voting by the people concerning both the revision/creation of laws and the election of politicians (Dunning 1909, p. 404). Furthermore, at least one scholar argues that though Rousseau may not be seen to be a "traditional" Liberal, he is most certainly promoting a revised version of Liberalism (cf. Sorenson 1990, pp. 443–66). Based upon the appearance, in the thought of Rousseau, of the three criteria of a Liberal Democracy, and the fact that we find him both promoting Democracy and a qualifed Liberalism, it seems that we are entitled to conclude that if he is not himself promoting a version of Liberal Democracy, then he is a legitimate forefather of Liberal Democratic theory. It is, indeed, as the legitimate forefather of a variety of contemporary Democracies that we find Rousseau presented to us by James Miller in his work on Rousseau's dream of democracy (Miller 1984, pp. 202–10).

70  Leclercq, *EP*, 158. Translated from the French, "Les hommes, abandonnant leur liberté à la communauté la retrouvent en elle, puisque la communauté, c'est eux; mais la communauté ne peut, à son tour, aliéner cette liberté. Le peuple ne peut se remettre à un souverai; il est lui-même le souverain, et le seul souverain possible".

71  John Rawls appears to agree that an actual separation of Church and State is not fully possible, but that it is possible to work around this through what he calls "overlapping consensus". In his paper, "The Idea of Overlapping Consensus", Rawls suggests a way of working around the fact that all of the members of any Democracy necessarily have a Religion or Life-Philosophy (what he calls a General and Comprehensive doctrine) which is the basis of their views about moral and political questions (Rawls 1987, pp. 9–11). His theory appears to state that, despite the pluralism of any contemporary Democratic state, there will be some rock-bottom ideas that all people agree upon, and that we can use these ideas as principles upon which to base the Democracy (Rawls 1987, p. 6). This may be a solution, though there are some potential difficulties with the theory proposed by Rawls, and it may yet fall prey to the problem we have underlined in this article. We do not, however, have space to consider his theory and its attending difficulties in this paper.

72  In the course of revising this paper for publication, we came across a paper published 71 years ago, by the eminent philosopher Gregory Vlastos, which appears to propose an argument which is very similar to the one we have articulted above. (cf. Vlastos 1953). Discussing Maritain's work *Man and the State*, Vlastos notes that Maritain clearly wants to defend both Democracy and the complete independence of Church and State (Vlastos 1953, p. 564), both functioning in their proper spheres (Vlastos 1953, pp. 565–66). Vlastos notes that the way that authority functions in the Church, versus in the State, is quite different (Vlastos 1953, pp. 566–67). Confirming our reading of Maritain, Vlastos notes that Maritain recognizes, as he must, that the two spheres (Church and State) are necessarily overlapping (Vlastos 1953, p. 569). Vlastos suggests, that the authority of the Church over its members in relation to moral matters brings it in to conflict with the State in relation to those same matters, thus creating precisely the problem we have raised above (Vlastos 1953, pp. 570–76). Though Vlastos approaches this subject from a slightly different direction, he appears to arrive at very similar conclusions.

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
