# Peer review of "Christian Citizens in a Democratic State: Is a True Separation of Church and State Really Possible?"

_religions, doi:10.3390/rel15030262_

Round 1

Reviewer 1 Report

Comments and Suggestions for Authors

This is an excellent, well-written, and cogently argued article which deserves to be published, contingent upon completing revisions.  Those revisions involve situating the author's claims in terms of other theories about the relationship between religion and politics, and in turn, toning down the article's somewhat strident language regarding its thesis. 

The article's main shortcoming is that, in the service of dramatizing its thesis--that a genuine separation between church and state is impossible--it begs the question in two main ways: the nature of democracy, and the meaningfulness, or value, of religious and moral actions.

To begin, the author describes democracy (or "true democracy") in terms which prearrange the conclusion that democratic states pursue given ends or goods.  As any student of contemporary liberal philosophy knows, however, this is an extremely debatable assertion, one which, to give just one very prominent example, John Rawls--who is not cited!--would firmly reject.  The nature of liberal democracy, for Rawls and like-thinkers, is precisely that it assumes agreement only about broad matters of political structure, not the deeper moral-metaphysical-ontological commitments (in his terms, "comprehensive doctrines") of its citizens.  Why isn't this, or other contemporary liberal theories, a "true" form of democracy?  In particular, for citizens who are not Christian, or religious, or have some kind of broadly Aristotelian conception of the Good, why would I necessarily conceive of the state in the teleological terms the author provides?

At the same time, the article operates on an unspoken assumption about what makes human acts valuable.  The author seems to assume that, in a situation in which the broader society is in engaging in acts deemed morally deficient in some way, it would be the state's proper role to attempt to reduce or end those acts, including through the use of force.  This, again, has to do with the author's broader assumption about the nature of political order.  However, if I'm Kant, say, or other deontological ethicist, or indeed a different kind of Christian, I might say that actions only acquire a genuinely moral character if they are performed with the right intention.  Going through the motions of the action itself has no moral value.  Consequently, it would be both wrong and counterproductive for the state to enforce morality on its citizens.  The author cites John Locke toward the beginning of the article, but does not engage with this basic Lockean (and Kantian) claim.

Author Response

Thank you for taking the time to read my article. I am quite encouraged by your comments concerning the value of the article. 

Though I cannot adequately integrate a response to every concern or question raised in your comments (without making the article unwieldy), I have attempted to provide clarity on the nature of Democracy, and the type of Democracy that is being considered (along with a brief overview of other forms of Democracy). I have included a discussion of the primary criteria of a liberal Democracy, and discussed some variations. I also added a footnote mentioning that there are a variety of approaches to church/state relations. Your second concern is interesting, however, I don't think I can adequately address it in this article. I tried to find ways to include some form of interaction with it in a footnote, but could find no proper context for that discussion. It might be an interesting subject for a follow-up article.

Reviewer 2 Report

Comments and Suggestions for Authors

Please see my Word comments!

Author Response

I was pleasantly surprised by the amount of attention that this reviewer gave to reading my article, and providing thoughtful responses. Thank you for your time and energy. There is a sense in which I could almost have seen a fruitful published discussion take place. On the other hand, I am happy to be able to read the comments and critiques prior to publication, and to thus improve my paper prior to publication.

I have attempted to respond to a number of the reviewers concerns in my paper, however, due to the amount of space which could have been dedicated to integrating all of them, I had to make some choices. Here is how I approached the reviewers comments: 

First, though I see no need to provide justification for my use of Maritain or Leclerq, I did provide a footnote which discusses the importance of Maritain for this debate. I was a bit surprised by the frequent call for "providing justification" in relation to Maritain, when others could have been used. I suspect that the reviewer may have put too much weight on the brief historical introduction to the paper. If I mentioned Baptist theologians, it was not to make this into a denominational discussion, but only to provide a brief historical prelude to a largely theoretical paper. It should be noted that after the historical prelude, I no longer deal with "historical" characters, but, rather, deal with the concepts/terms. Indeed, I made it clear at the beginning that I am actually using a contemporary theorists interpretation of the concept of absolute separation as the basic claim (I noted that I am taking the interpretation of Derek Davis as the basic understanding of the "wall of separation"). To clarify that I am not dealing with just Baptist political theory, or historical political theory, I added in a footnote pointing to a prominent American Lawyer (whose religious affiliations are unknown, to me at least), who proves an approach to separation of Church and State which, though not precisely the same as that proposed by Derek Davis, still falls prey to my argument). The purpose of this paper, as stated at the outset, is not to refute Baptist political theology, but to show that an Absolute Separation of Church and State, as proposed by some Baptist Political theorists (among others), is impossible in a "true democracy". For that purpose, as the reviewer appears to agree, Maritain's thought is quite helpful. I noted this in a footnote, and added an additional statement to make it very clear precisely what it being done in this paper. Though the purpose of this paper is not to criticize a specifically Baptist perspective, I still see no reason why I would need to provide justification for the use of a Catholic in criticizing a Baptist theologian. This is, in fact, quite normal in academic papers and presentations. Some of the greatest Catholic scholars appeal to Protestant authors in very much the same way as I have here; some of the greatest Protestant scholars appeal to Catholic scholars in the same way. In fact, during the Reformation, it was commonplace for Reformers to appeal to Roman Catholic theologians in order to refute other Roman Catholic theologians (and vice versa).
I thought the third point (using Maritain, a defender of Democracy and the separation of church and state, to critique the separation of church and state) was helpful, and added a footnote explaining why, and pointing to relevant secondary literature. It is worth noting, even Maritain appears to have recognized that an absolute separation of church and state was not possible (see footnote). As such, he is in fact an ally to my argument, rather than an enemy.

To the fourth and fifth comments, this seems to be a comment on other ways in which the article could have been written. I agree, I could have gone about it different ways. I don't see this as a valid critique of how I actually did go about it.

The sixth comment was helpful. It is very true that there are other ways of defining the separation of church and state, as well as other ways of working out what separation of church and state entails. I was clear that I am not working with all versions, but with a specific type. I added a footnote to note that there are other versions. I mention and provide a quick summary of another contemporary version. I also note that I think that this version also falls prey to my argument. I doubt that more can be said, as the purpose of this paper (as stated) is not to criticize every possible approach to the separation of church and state, but to criticize one prominent (and actually widely accepted) understanding of the separation of church and state. The seventh comment appears to be similar in intent to the 6th. I think that I have added sufficient nuances to allay the concerns expressed here.

Finally, one other concern raised by the reviewer was the nature of Rousseau's approach to politics. I have added a footnote to explain the attribution of Democracy to Rousseau, and to point out that there is debate about the nature of Rousseau's approach to democracy. I don't know if it was necessary to add as much detail on this question as I ended up adding, but, it will at least be clear.

Once again, I very much appreciate the deep reading this reviewer gave to my paper, and think that the paper has been improved by my attempts to integrate responses to the comments they provided.

Reviewer 3 Report

Comments and Suggestions for Authors

Overall, I find this research to be very interesting.  While this subject is widely recognized and frequently debated, you've added your own unique touch to it. I commend the researchers for looking at this thopic.

The only issue I have with this paper is the way references are cited. Nevertheless, if the editors recommended the use of the Chicago citation style, then everything is acceptable in the paper.

Author Response

I appreciate the reviewers kind comments. Thank you.

Round 2

Reviewer 2 Report

Comments and Suggestions for Authors

Good evening,

Here, I'll leave the same comments for the author and the editors--I just cannot sign off on this piece in its current form.

First, this was a rejection (with encouragement to resubmit).  Rejections are not handled by modifying footnotes; much more was needed here.  I submitted close to 3,000 words justifying my rejection, and by the author's own admission not all the points have been addressed.  This is an indication to me that something is off.  It is not my job, or that of any reviewer, to make the author comfortable.  It is the author's job to make the reviewer comfortable, by responding to all points, even ones with which the author disagrees.  The general annoyance on the part of the author, in some of the footnotes and in the response letter, with having to provide justification for theoretical moves made by the article, further reinforces to me that there is an issue here with understanding the overall aim and purpose of peer review.

One of the important points that has not been addressed is that this is a straw man argument: which Baptist or Protestant thinker ever advocated for the "total separation of church and state," which would involve a person not bringing any part of their religious identity into the public square?  Certainly not Roger Williams.  So, then, if Derek Davis claims that Williams said anything like this, he is wrong; and if Joseph Dawson is arguing as much, he, too, is wrong.  The author may disagree, but then he or she needs to acknowledge recent research (I mentioned Teresa Bejan's book on civility and Williams as just one example) that clearly shows Williams favoring expression that could include religious reasons in public.

If there are other actual examples of low-church Protestants who advocated for a total separation of church and state as defined by this article, please provide them.

If you can't provide them, you need to mention some of the recent research that successfully contests the total separationist view of evangelicals like Williams, and you can't make unsupported claims in the abstract along the lines of, "In many North American Protestant circles, especially those with Baptist or Free Church roots, the notion of the total separation of Church and State is presented as the ideal to be attained in all Church and State relations." 

It sounds as if the real issue is with the early Rawls and his insistence on "public reason."  I wholeheartedly agree--but then no North American Protestant straw men, please.  You open the piece with them, and not with Davis or Dawson, who are never discussed in the text of the article.

Same goes for Rousseau.  You do need to provide a justification for using him as a theorist of liberal democracy ... given that he was patently not a theorist of liberal democracy.  And this justification needs to go into the text, not a footnote.  Of course, it is well known that in academia it is possible to find a source, somewhere, saying just about anything; here, in the text of the article, you need to provide reasons for going with a minority interpretation of Rousseau, given that the majority interpretation (of him as a republican who saw individual rights, at best, as secondary) makes this a tautological argument.  Of course Rousseau would have opposed the "total separation of church and state"--he was a republican (not a liberal democrat), and prescribed a civil religion in line what he and all other classical republicans saw as the necessary continuity between religion and politics!

Looking forward to seeing all concerns addressed.

Author Response

So, in response to the reviewer, I would repeat that my paper does not hinge upon the specific views of Roger Williams or Isaac Backus. They are mentioned as historical figures in the development of what would become known as the "wall of separation". Here is the sentence in which Williams is mentioned: "Early English Puritans, such as Roger Williams (c. 1603-1683), and early American Baptists such as Isaac Backus (1724-1806)—who was influenced by the likes of Williams and the English philosopher John Locke (1632-1704), argued energetically for religious freedom." I have not attributed to Williams or Backus, the view of separation of church and state which I call "absolute separation". I say, rightly, that they "argued energetically for religious freedom." Aside from a few footnotes (where I point to how other scholars have read the notion of the "wall of separation" in articles related to Williams), this is the only time where Williams is mentioned in relation to church state relations. As I stated in my previous response, and have re-iterated here, Williams is introduced in this paper as a historical influence on the American discussion about church-state relations. The reviewer may very well be right about how Williams thought about Church-state. This is entirely immaterial to both what I have said about Williams, and to the purpose of this paper. I leave it to the editors of the journal to decide whether the reviewer's insistence is grounded, and I will make any edits that the editors think are necessary.

The reviewer suggests that I have not provided resources which show that some scholars are advocating for the form of absolute separation that I am interacting with. This has, in fact, been done: (1) by referring to Derek H. Davis, who appears to suggest that the Wall of Separation be read this way, with attending footnotes. cf. fn. 3-7. (2) I have also noted, cf fn. 8, that there are other ways of interpreting the "wall of separation", which contain a more nuanced approach, but still fall prey to the argument presented in my paper. (3) I have also noted, finally, cf. fn. 2, that there are other ways of interpreting Jefferson's wall of separation. Now, fn. 8 was added in after the first reviews. Between the footnotes 2-8, I have provided ample evidence of specifically the type of absolute separation of church and state which my paper is seeking to argue against. I leave it to the editors to decide if I have answered this reviewers critique on this question. 

The reviewer says that I have committed a straw man fallacy by attributing to early Baptists (I assume he is again referring to Williams and Backus) a view they didn't hold, and that "You open the piece with them, and not with Davis or Dawson, who are never discussed in the text of the article." Davis's views have been explicitly mentioned, from the very first draft of this paper, in the second paragraph of the manuscript (cf. pg. 2). Not only are they discussed in the body of the text, as I explain the approach to Church-State relations that my article is interacting with, they are also brought up again, later in the article. I leave it to the editor to make the final decision about whether they need to be mentioned more than they already are, and to decide if I am strawmanning early Baptist theologians (see my first response above).

The reviewer again mentions the fact that I have mentioned Rousseau, and contests the fact that Jean-Jacques Rousseau is said to be a "liberal democrat". Now, first of all, the reviewer is right to note that in the first draft of my paper, I could be quoted as saying, "the liberal democracy of philosophers like Jean-Jacques Rousseau". However, after the comments of the reviewers, I changed the sentence to say, rightly, "A similar argument can be found in the writings of Jacques Leclercq, as he is interacting with what he qualifies as a version of liberal democracy, which he sees as originating in the thought of philosophers such as Jean-Jacques Rousseau." Now, as I note in the attached footnote (cf. fn. 70) the reviewer is probably right to say that Rousseau was not a "liberal democrat" (at least not according to contemporary standards), however, as I also note in my footnote, up until around the 1930s it was very commonplace for scholars to see Rousseau as a Liberal Democrat. Jacques Leclerq certainly thought that Rousseau was a Liberal Democrat, and this is what I say in the body of my text. I also note, in the same footnote, that there is an academic debate about whether or not Rousseau was, himself, a Liberal and a Democrat. The reviewer clearly takes one side of the debate. That is fine, and the reviewer has the right to do that. However, the reviewer should at least be willing to recognize that there is academic debate on the issue (and a number of published books and articles have noted the reasons why there is a debate). I point out that, based upon the most basic criteria for adhering to liberal democracy, if Rousseau did not adhere to this form of governance, he most certainly was one of its forefathers. Now, other articles and books could be (and have been) written debating the question of Rousseau's political leanings. However, what I specifically say in my article is that Leclerq presents helpful comments in the context of his discussion of Liberal Democracy, "which he sees as originating in the thought of philosophers such as Jean-Jacques Rousseau." This statement is a true statement, as it is not describing Rousseau, but Leclerq's reading of Rousseau. As such, I do not need to provide justification for "using him as a theorist of liberal democracy", as I am not using him as a theorist of liberal democracy. That I am not using him as a theorist of liberal democracy is clear in the second draft which was sent after the first series of reviews, and to which this reviewer is here responding. In the second draft, I addressed this question both in the body of the text (as the quote I provided above shows), and added clarifying remarks in the attached footnote. It is unclear to me why this reviewer makes the comment they made, as they are clearly not interacting with what I said in the second draft of my paper (with which their second review is supposed to be interacting). Again, I leave this to the editors to decide if more clarification needs to be made.

I am sorry if the reviewer felt that my responses were inadequate. I responded to every single point that they brought up (sometimes noting that my response to one of their points served to respond to other points they brought up, such that it was unnecessary to repeat myself). Again, I leave it to the editors to decide if more needs to be done to adequately respond to the reviewer.